# Analysis of Polyphenolic Compounds in Water-Based Extracts of *Vicia faba* L.: A Potential Innovative Source of Nutraceutical Ingredients

**DOI:** 10.3390/antiox11122453

**Published:** 2022-12-12

**Authors:** Luigi Castaldo, Luana Izzo, Sonia Lombardi, Anna Gaspari, Stefania De Pascale, Michela Grosso, Alberto Ritieni

**Affiliations:** 1Department of Pharmacy, University of Naples “Federico II”, 49 Via Domenico Montesano, 80131 Naples, Italy; 2Department of Agricultural Sciences, University of Naples “Federico II”, 80055 Portici, Italy; 3Department of Molecular Medicine and Medical Biotechnology, School of Medicine, University of Naples “Federico II”, 5 Via Sergio Pansini, 80131 Naples, Italy

**Keywords:** bioactive compounds, orbitrap, polyphenols, agro-waste valorization

## Abstract

The water-based extract of broad bean hulls contains several bioactive molecules, including polyphenols well-known to exert antioxidant activity, which could justify its use in nutraceutical formulations. Hence, the current investigation aimed to establish the polyphenolic profile of water-based extracts from broad bean hulls through UHPLC–Q-Orbitrap HRMS analysis. The findings highlighted that *p*-coumaric acid, chlorogenic acid, and epicatechin were the most common compounds found in the tested extracts, being quantified at a mean concentration of 42.1, 32.6, and 31.2 mg/100 g, respectively. Moreover, broad bean hull extracts were encapsulated into a nutraceutical formulation, after which the antioxidant properties and the bioaccessibility of phenolic compounds during the simulated gastrointestinal (GI) process were investigated and compared with the digested non-encapsulated extract. The data highlighted that following the GI process, the capsules were able to preserve active compounds from the adverse effects of digestion, resulting in a greater antioxidant capacity and polyphenol bioaccessibility in the duodenal and colonic phases, compared with the non-encapsulated extract. Our results showed that the water extract from broad bean hulls may be considered a valuable source of natural polyphenolic compounds; in addition, the use of a gastric-resistant capsule could be a suitable alternative to transport these bioactive compounds to the target tissues.

## 1. Introduction

*Vicia faba* L., commonly called broad bean, is an early legume crop native to Middle Eastern countries that has become extensively naturalized in several other regions around the world [1]. Broad bean seeds have been consumed for a long time for human and animal nutrition due to their low cost and ease of growing, as well as their great nutritional relevance which includes a high amount of protein, fiber, carbohydrates, vitamins, and minerals [2]. However, anti-nutritional factors have been found in broad bean seeds including phytic acid, which has been linked to the reduced absorption of several minerals [3].

As reported by the Food and Agriculture Organization (FAO), 5.4 million metric tons of broad beans were produced globally in 2020 [4]. China is the greater producer in the world with 1.7 million metric tons annually, whereas Europe produces 30% of the world’s yearly production. The usual consumption of broad beans involves the removal of the hulls. As a result, the broad bean industry generates a significant amount of agro-waste, which poses a serious problem in terms of environmental impact and the loss of valuable nutrients. Typically, agro-waste contains several bioactive molecules that could be employed effectively in the formulation of nutraceutical products [5]. Nowadays, the use of agricultural waste and by-products is spreading as it is a novel responsible way to reduce adverse environmental effects and encourage sustainable development. Overall, the most environmentally friendly approaches to recover active compounds from agro-waste and minimize environmental impact are typically water-based extraction procedures [6,7]. According to studies, around 70% of solid waste in the broad bean industry is made up of broad bean hulls [8]. To date, only a minimum part of the broad bean hulls are employed as animal feed, and despite efforts to find a different application for these by-products including the production of biofuels and biogas [9], large amounts of broad bean hulls are left in the field.

Some scientific data report that broad bean hulls could contain active substances including polyphenols [10]. Polyphenols are a class of phytochemicals found in several plant-based foods including tea, fruits, vegetables, and nuts, among others [11,12]. A growing amount of data indicates that polyphenol-rich diets can help sustain human health by preventing a variety of age-related diseases [13], and most of the main health effects ascribed to polyphenols are due to their well-known antioxidant activity [14]. However, limited literature is currently available about the polyphenolic composition of broad bean hulls. A recent report [15] identified some important phenolic and flavonoid compounds in broad bean hull extracts using high-performance liquid chromatography (HPLC) with a diode-array detector (DAD). Further scientific studies have reported important pharmacological properties for these active substances, which could justify the exploitation of broad bean hull extracts as a novel source of active molecules [16,17].

In recent decades, the methods mainly employed for the detection of polyphenols in plant-based foodstuffs are based on the LC technique coupled with mass spectrometry (MS) technologies [18,19]. The use of Q-Orbitrap, a high-resolution mass spectrometer (HRMS), in combination with ultra-HPLC is well-known as an optimal way for detecting a variety of active substances in vegetal foods due to its high specificity and sensitivity, which enables quantification based on precise mass measurement [20].

Several studies have highlighted that the absorption of active compounds, including polyphenols, occurs mostly in the colonic stage after being metabolized by intestinal microbiota [21]. Scientific studies report that during the gastrointestinal (GI) process, polyphenols are particularly susceptible to the activity of digestive enzymes, intestinal microflora, pH, and temperature, which may affect their bioaccessibility [22]. Different potential effective methods have been suggested to overcome this limitation. The use of nutraceutical formulations could effectively improve the bioaccessibility of polyphenols, preserving their chemical structure which may be influenced during the GI process, as recently reported [23].

Therefore, the current investigation aimed to establish the polyphenolic profile of water-based extracts from broad bean hulls through UHPLC–Q-Orbitrap HRMS analysis. Moreover, broad bean hull extracts were encapsulated into a nutraceutical formulation, after which the antioxidant properties and the bioaccessibility of phenolic compounds of broad bean hull after in vitro GI digestion were investigated, in order to promote the exploitation of this novel source of antioxidants in the formulation of nutraceuticals.

## 2. Materials and Methods

### 2.1. Sampling

The broad bean (*Vicia faba* L.) samples used in the present study were collected in May 2022 from various locations in Southern Italy (Campania region). The broad bean seeds were manually removed from the hulls. After cutting the green hulls into little pieces, they were lyophilized to a fine powder and kept at 4 °C until the polyphenol extraction procedure.

### 2.2. Reagents and Materials

The deionized water used for the sample extraction was purchased from Millipore (Bedford, MA, USA), whereas the methanol, formic acid (FA), and water (LC–MS grade) used for the chromatography were provided by Carlo Erba reagents (Milan, Italy). The polyphenol standards (purity > 98%), namely protocatechuic acid, gallic acid, quinic acid, apigenin-7-O-glucoside, *p*-coumaric acid, catechin, epicatechin, chlorogenic acid, rosamarinic acid, hesperidin, quercetin-3-galattoside, rutin, naringin, ferulic acid, apigenin, luteolin, gallic acid, naringenin, quercetin, daidzein, myricetin, isorhamnetin-3-rutinoside, genistein, diosmin, and ellagic acid kaempferol-3-glucoside were purchased from Sigma-Aldrich (Milan, Italy). In order to perform antioxidant tests, hydrochloric acid, 1,1-diphenyl-2-picrylhydrazyl (DPPH), potassium persulphate, 6-hydroxy-2,5,7,8-tetramethylchromane-2-carboxylic acid (Trolox), and 2,2′-azino-bis-3-ethylbenzthiazoline-6-sulphonic acid (ABTS) were provided by Sigma-Aldrich (Milan, Italy). The enzymes and standards used in the simulated GI, namely pancreatin, α-amylase (1000–3000 U/mg solid) from human saliva, bile salt, pepsin (≥2500 U/mg solid) from porcine gastric mucosa, Pronase, and Viscozyme L., were purchased from Sigma-Aldrich (Milan, Italy).

### 2.3. Extraction of Polyphenols

Water-based extracts from broad bean hulls were obtained following the previously described protocol [24]. In short, 200 mL of heated (80 °C) water was combined with 10 g of pulverized material. Then, the mixture was centrifuged at 4900× *g* for 5 min after stirring at 300× *g* for 15 min at 80 °C in a shaking water bath (IKA Basic KS 130, Argo Lab, Rodano, Italy). Afterward, the supernatant was recovered, while the residual pellet was extracted again following the same procedure. Then, the obtained supernatants were pooled before being frozen and lyophilized. Finally, 0.500 g of polyphenolic extract of bean hulls was encapsulated in acid-resistant capsules of hydroxypropyl methylcellulose (pharmaceutical grade), whereas 0.500 g of cellulose was used to replace the polyphenolic extract of bean hulls in the control capsules (CP).

### 2.4. Phytic Acid Concentration

The phytic acid content of the broad bean hull water-based extract was evaluated according to the procedure proposed by McKie [25] using a commercial phytic acid assay kit (Megazyme Ltd., Bray, Ireland). In short, 0.66 M HCl solution was used to extract phytic acid. Afterward, multiple enzymatic processes were performed to release inorganic phosphorus, after which the solution was treated with ammonium molybdate. The absorbance at 655 nm of the final mixture, which contained molybdenum blue, was used to calculate the phytic acid concentration. The results were displayed in mg of phytic acid per 100 g of broad bean hull water-based extract.

### 2.5. UHPLC and Orbitrap HRMS Analysis

The polyphenolic components were separated chromatographically using an UHPLC device (Dionex UltiMate 3000, Thermo Fisher Scientific, Waltham, MA, USA) equipped with an autosampler, a quaternary UHPLC pump, and a degassing system. A thermostated (25 °C) Kinetex F5 column (50 × 2.1 mm, 1.7 µm, Phenomenex, Torrance, CA, USA) was used for chromatography separation. The eluent phases were water (A) and methanol (B), both containing 0.1% FA. The gradient elution was: 0–0.5 min, 100% A; 0.5–1 min, 100–60% A; 1–2 min, 60–20% A; 2–5 min, 20–0% A; 5–9 min, 0% A; 9–11 min, 0–100% A; 11–13 min, 100% for column re-equilibration. The injection volume was set at 5 µL, while the flow rate was set at 0.5 mL/min.

A Q-Exactive Orbitrap mass spectrometer (Thermo Fisher Scientific, Waltham, MA, USA) in negative mode was used for detection. Full ion MS data were acquired at a scan range 80–1000 *m/z*, auxiliary gas heater temperature 350 °C, S-lens RF level 60, capillary temperature 320 °C, spray voltage 3.5 KV, sweep gas flow rate 0, auxiliary gas 3, sheath gas flow rate 18, maximum injection time 200 ms, AGC target 1 × 10^6^, microscan 1, and resolution power of 70,000 FWHM. Detection was achieved by setting the mass tolerance to 5 ppm. Xcalibur software 3.1.66.19 (Thermo Fisher Scientific, Waltham, MA, USA) was used to process the data.

### 2.6. In Vitro GI Process

The in vitro GI process was performed following the protocol proposed by the INFOGEST network [26] to assess the variation in antioxidant activity and polyphenol bioaccessibility during the various stages of the GI process. The simulated solutions, namely salivary fluid, gastric fluid, and intestinal fluid (SSF, SGF, and SIF), were prepared following the previously reported salt concentrations [26] (Appendix A, Appendix A).

Broad bean hull water-based extracts, both encapsulated and non-encapsulated, were suspended in 0.5 mL of α-amylase solution, 25 µL of 0.3 M calcium chloride, 3.5 mL of SSF, and 975 µL of H_2_O. Before incubating for 30 s at 37 °C, the pH of the samples was adjusted to 7. Afterward, the gastric phase was simulated by the addition of 0.69 mL of water, 5 μL of 0.3 M calcium chloride, 1.6 mL of a pepsin solution, and 7.5 mL of SGF to the mixture. Before incubating for 2 h at 37 °C, the pH of the samples was adjusted to 3.

Then, the intestinal phase was simulated by the addition of 1.3 mL of water, 40 µL of 0.3 M calcium chloride, 11 mL of SIF, 2.5 mL bile salt solution, and 5 mL pancreatin solution to the mixture. Moreover, the mixtures were incubated for 2 h at 37 °C. The pH was rose to 7 before the incubation step. At the end of all GI phases, all supernatants were collected and lyophilized. Moreover, to simulate the colonic step, the remaining pellets obtained after the intestinal phase were treated using the protocol previously reported [27]. In short, to the remaining pellets, Pronase solution (5 mL, 1 mg/mL) was added, and incubated for 60 min (pH 8). Afterward, 5 mL of H_2_O and 150 µL of Viscozyme L. were added to the mixture. Finally, the samples were incubated for 16 h at 37 °C (pH 4) and then centrifuged. The supernatants were recovered and lyophilized.

### 2.7. Antioxidant Activity

The antioxidant activity of water-based extracts from digested and non-digested bean hulls was evaluated using two different assays: DPPH and ABTS. The data were displayed as mmol per kg of extract.

#### 2.7.1. DPPH Test

The DPPH test was performed following a previously published protocol [28]. In brief, a DPPH standard (1 mg) was diluted with methanol until the absorbance reached 0.90 ± 0.02 at 517 nm. Afterward, 0.2 mL of each sample was added to 1 mL of DPPH working solution (WS). Finally, after 10 min, the absorbance was monitored at 517 nm.

#### 2.7.2. ABTS Test

The ABTS test was performed based on the protocol reported by Izzo et al. [29]. In brief, 2.5 mL of aqueous ABTS (7 mM) was added to potassium persulfate (44 µL, 2.45 mM). After incubation at room temperature for 16 h, the ABTS mixture was diluted with EtOH until the absorbance reached 0.70 ± 0.02 at 734 nm. Finally, the analysis was carried out by adding 1 mL of ABTS WS to 100 μL of the sample, and after 3 min, the absorbance was monitored at 734 nm.

### 2.8. Quantification of Total Phenolic Content

The total phenolic content (TPC) was determined following a previously published procedure [30]. In short, 125 µL of the Folin–Ciocalteu reagent (2 N) was added to 125 µL of sample and 0.5 mL of water. After incubation at room temperature for 6 min, 1 mL of water and 1.25 mL of sodium carbonate solution (7.5%) were added to the mixture. Finally, the absorbance was recorded after 60 min at 760 nm.

### 2.9. Statistical Analysis

Tukey’s test was used to determine the significance of differences between the means at the level of significant *p*-values of less than 0.05. Using Pearson’s method, the correlation coefficients were assessed. Stata 12 software was used for data processing (StataCorp LP, College Station, TX, USA).

## 3. Results

### 3.1. Phytic Acid Concentration

The concentration of phytic acid in broad bean hull water-based extracts was assessed using an enzymatic test. Phytic acid levels in the tested samples ranged from 46.2 to 51.7 mg/100 g of sample, with a mean content of 49.3 mg/100 g of sample.

### 3.2. Identification of Flavonoids and Phenolic Compounds in the Broad Bean Water-Based Extracts

In order to detect active compounds (*n* = 22), such as phenolic acids (*n* = 6) and flavonoids (*n* = 16) in broad bean hull extracts, an UHPLC–Q-Orbitrap HRMS investigation was carried out. The data demonstrated that using the UHPLC system, the examined compounds were successfully separated over a 9-min run time. Table 1 shows the mass parameters such as sensitivity, accuracy, measured mass and theoretical (*m*/*z*), chemical formula, retention time (RT), and ion assignment. Experiments were achieved in negative ESI- mode, and the data were monitored in full-scan HRMS. The structural isomers genistein and apigenin-7-O-glucoside (*m*/*z* 269.04554), and hesperidin and rutin (*m*/*z* 609.14611) were confirmed by comparing the retention times of the studied compounds with those of real standards and with data reported in previously published articles.

### 3.3. Quantification of Flavonoids and Phenolic Compounds in the Broad Bean Water-Based Extracts

The main flavonoids and phenolic acids found in the water-based extracts of broad bean hull extracts were quantified through a UHPLC–Q-Orbitrap HRMS analysis. All the studied analytes were quantitatively analyzed using calibration curves built in triplicate at eight different concentration levels. Regression coefficients greater than 0.990 were found. As shown in Table 2, the total concentration of phenolic acids found in the water-based extracts of broad bean hulls reached levels of up to 103 mg/100 g. The findings highlighted that roughly 52% of all the polyphenolic compounds in the tested samples were phenolic acids. In addition, the most abundant phenolic acid found in the water-based extracts of broad bean hulls was *p*-coumaric acid (*p*-value ≤ 0.05), with an average value of 42.1 mg/100 g.

Moreover, the levels of some important flavonoids, namely isoflavone, flavonols, flavanones, flavanols, and flavones, were quantified in assayed extracts. As far as flavanols were concerned, epicatechin and catechin were the most relevant flavonoids (*p*-value ≤ 0.05) and showed a total concentration of 43.3 mg/100 g, making up 21.8% of the total polyphenols found in the water-based extracts of broad bean hulls. Flavanones, mainly represented by hesperidin and naringenin, were found with an average value of 15.4 and 2.8 mg/100 g, respectively. Flavanones represented 9.2% of the total polyphenols found in the assayed extracts. Regarding the levels of flavonols found in water-based extracts of broad bean hulls, rutin was the most abundant analyte (*p*-value ≤ 0.05), with an average value of 12.3 mg/100 g. However, none of the following analytes were found in the tested samples: daidzein, naringin, isorhamnetin 3-rutinoside, apigenin-7-O-glucoside, or protocatechuic acid.

### 3.4. Bioaccessibility of Polyphenols in Water-Based Extract of Broad Bean Encapsulated and Non-Encapsulated

The in vitro simulated GI process was performed on the encapsulated and non-encapsulated water-based broad bean extract to obtain information on the ability of the capsules to protect polyphenol compounds during the GI process. The Folin–Ciocalteu method was used to measure TPCs in the small intestinal and colon stages. The total value of the colonic stage was calculated by summing the data of the Viscozyme L and Pronase phases. In comparison to the TPC value found in the broad bean extract, the data showed that the TPC levels in both encapsulated and non-encapsulated extracts significantly decreased (*p*-value ≤ 0.05) after the GI digestion phases, as evidenced in Table 3. Furthermore, comparing the encapsulated and non-encapsulated extracts after the small intestinal and colonic stages, the data showed that the encapsulated extracts had significantly higher TPC values (*p*-value ≤ 0.05) than the non-encapsulated extracts. The TPCs observed in the total colonic phase in the encapsulated and non-encapsulated extracts were quantified at an average value of 8.2 mg GAE/g and 6.6 mg GAE/g, respectively. After the colonic phase, both the encapsulated and non-encapsulated extracts showed the greatest TPC levels (*p*-value ≤ 0.05). Concerning the CT samples, Appendix A (Appendix A) shows the results recorded after the simulated GI process.

### 3.5. Antioxidant Activity of Water-Based Extract of Broad Bean Encapsulated and Non-Encapsulated

The antioxidant properties of the encapsulated and non-encapsulated water-based broad bean hull extracts were measured by using two different tests, namely DPPH and ABTS, after the intestinal, Pronase, and Viscozyme L phases, in order to highlight the protective effect carried out by the nutraceutical form against the action of the GI process. The antioxidant properties of both encapsulated and non-encapsulated extracts significantly decreased after all stages of GI digestion in all assayed methods (DPPH and ABTS tests) compared with the non-digested ones, as shown in Table 4. The overall colonic stage value was considered as the sum of the data from the Pronase and Viscozyme L phases. The results highlighted that, in all conducted spectrophotometric assays, the higher antioxidant properties of the encapsulated extract were revealed when monitored during simulated GI compared with the non-encapsulated extract in both the intestinal and colonic stages. The results obtained from DPPH and ABTS assays were correlated with the TPC data acquired during the in vitro GI process, as shown in Appendix A (Appendix A). Regarding the CT, Appendix A (Appendix A) shows the data monitored after the simulated GI process.

## 4. Discussion

The main objective of this study was to add value to broad bean hulls, a waste product derived from the agro-industry, by providing relevant information on the active ingredients, in particular the polyphenols, which are present in this agro-waste. An aqueous-based extraction procedure was carried out to minimize the environmental impact and obtain a food-grade material from broad bean hull samples. The findings highlighted a low concentration of phytic acid in the assayed water-based extracts from broad bean hulls. Although previous data have shown that phytic acid consumption is related to harmful effects [31], some studies have also suggested that consuming phytic acids at low levels has advantages for human health [32].

Overall, the results of the present study indicate that broad bean hull material may be a potential alternative source of polyphenolic substances, such as chlorogenic acid, coumaric acid, ferulic acid, catechin, and epicatechin, as well as many other significant phenolic compounds. Broad bean hull water-based extracts showed a total phenolic acid concentration of 103.0 mg/100 g of sample. Moreover, the levels of some important flavonoids, including flavonols, isoflavone, flavanones, flavanols, and flavones, were quantified in the assayed extracts (total mean of 95.2 mg/100 g). Although the polyphenolic profile of broad beans has been extensively investigated, the study of polyphenols has barely been evaluated in their hulls. To our knowledge, this report is the first work that evaluates the polyphenolic profile of an aqueous-based extract of broad bean hulls through the UHPLC–Q-Orbitrap HRMS technique. In fact, in previous works, the extraction of polyphenols from broad bean hulls was performed using different solvents including ethanol, ethanol–water, acetone, methanol, butanol, and ethyl acetate, while a simple aqueous-based procedure capable of recovering bioactive compounds that could be used in the nutraceutical formulation had not been studied. Recently, Valente et al. [8] investigated the phenolic profile of methanolic extracts from broad bean hulls by the HPLC–DAD–MS/MS method, and the findings highlighted that the predominant phenolic compounds present in this extract were coumaric acid and caffeic acid. Similarly, Ceramella et al. [15] studied the polyphenolic fraction of broad bean hulls extracted with three methods: methanol, 70% aqueous ethanol, and acetone, reporting remarkable amounts of chlorogenic acid, coumaric acid, and ellagic acid. The TPC value found in the analyzed water-based extracts was lower than those reported by Hashemi et al. [33], who obtained methanol-based extracts of broad bean hulls with Soxhlet, percolation, and ultrasound-assisted extraction, which reported TPC values ranging from 54 to 110 mg GAE/g. Moreover, in our previous investigation performed on the water-based extract of broad pea hulls [30], we reported that pea sample extracts showed a lower TPC value (9.7 vs. 8.2 mg/g). Furthermore, the two most significant compounds identified in the pea sample extracts were chlorogenic acid and epicatechin, with average values of 59.87 and 29.46 mg/100 g, respectively.

Although numerous scientific studies suggest that a regular dietary intake of polyphenols may play a critical role in maintaining human health and preventing a number of non-communicable diseases, it is fundamental to take into consideration that these molecules are extremely sensitive, and the GI process could affect the structure as well as the biological properties of these active metabolites, restricting their bioaccessibility. In consideration of this, broad bean hull extracts were encapsulated into a nutraceutical formulation and an in vitro simulated GI process was performed on the encapsulated and non-encapsulated extract to obtain information on the ability of the acid-resistant capsules to protect polyphenol compounds during the GI process. In the present scientific work, in order to mimic the effect of oral, gastric, and intestinal digestion, the INFOGEST protocol was performed [26]. This procedure is widely regarded as one of the most effective protocols for simulating the natural digestion process. In vitro intestinal models represent the gold standard in such investigations; in fact, they can rapidly provide useful information on the impact of food components on health status [34]. Despite the fecal inoculum representing the most appropriate protocol to replicate in vitro colonic digestion, an increasing number of studies have reported that the combination of bacterial enzymes, such as Pronase and Viscozyme L, represent a suitable alternative to reproduce intestinal fermentation [23,24,35,36,37,38]. Several methods have been proposed to evaluate antioxidant capacity after simulated GI digestion. The use of immortalized cell lines such as Caco-2 represents the most common technique for assessing antioxidant capacity and polyphenol bioaccessibility [39]. However, to obtain an overview of antioxidant capacity in the present work, DPPH and ABTS tests were performed after the GI process. The data highlighted strong correlations among the results obtained from TPC by the Folin–Ciocalteu and antioxidant tests assessed during the in vitro GI process, supporting that the assayed methods provide some reliable information on the antioxidant compounds released by the nutraceutical formulations assayed during the simulated GI digestion. Our findings showed that in both the intestinal and colonic phases, the bioaccessibility of the polyphenols and the antioxidant activity of the broad bean hull extract contained in acid-resistant capsules were higher compared with the non-encapsulated extract. Our results are in agreement with those reported by Izzo et al. [23], who highlighted that encapsulated red cabbage extract after simulated GI showed a higher intestinal and colonic bioaccessibility compared with the non-encapsulated extract. Similarly, Amrani-Allalou et al. [40] highlighted that the encapsulated extract of P. *spinosa*, a novel source of polyphenols, exhibited a higher TPC value following GI digestion than the same non-encapsulated extract. Our results showed that the use of acid-resistant capsules was able to preserve the polyphenol compounds from the GI environment, protecting their chemical structure, as well as their antioxidant capacity. Therefore, using a capsule formulation could be considered as a valid strategy in antioxidant delivery to target tissues, thus exerting a more effective health-promoting activity.

## 5. Conclusions

In summary, an in-depth investigation of the polyphenolic profile of the water-based extract of broad bean hulls using UHPLC–Q-Orbitrap is provided for the first time by this study. Within this extract, *p*-coumaric acid, chlorogenic acid, and epicatechin were the three most commonly found polyphenolic compounds in the assayed extracts, being quantified at mean values of 42.1, 32.6, and 31.2 mg/100 g, respectively. Furthermore, our data showed that following the GI process, the use of capsules was able to preserve the active compounds found in the extracts from the adverse effects of digestion, resulting in a greater antioxidant capacity and TPC values in the duodenal, as well as colonic, phases compared with the non-encapsulated extract. Our findings demonstrate that water-based extracts from broad bean hulls could be considered a valuable source of natural polyphenolic compounds; in addition, the use of gastric-resistant capsules could be a valid alternative to transport these bioactive compounds to the target tissues. However, further research is needed to expand our understanding regarding the bioactivity of the studied extracts for possible applications as ingredients in nutraceutical formulations, and to elucidate the biotransformation of the polyphenolic compounds that occurs during the GI process.

## Figures and Tables

**Table 1 antioxidants-11-02453-t001:** UHPLC–HRMS parameters of the investigated analytes.

Compound	RT (min)	Adduct Ion	Chemical Formula	Theoretical Mass (*m*/*z*)	Measured Mass (*m*/*z*)	Accuracy (Δ mg/kg)	LOD (mg/kg)	LOQ (mg/kg)
Quinic acid	0.47	[M-H]^−^	C_7_H_12_O_6_	191.05531	191.05611	4.18727	0.019	0.057
Gallic acid	0.82	[M-H]^−^	C_7_H_6_O_5_	169.01425	169.0149	3.84583	0.039	0.117
Protocatechuic acid	1.60	[M-H]^−^	C_7_H_6_O_4_	153.0193	153.01857	−4.77064	0.019	0.057
Epicatechin	3.09	[M-H]^−^	C_15_H_14_O_7_	289.07176	289.07202	0.89943	0.019	0.057
Chlorogenic acid	3.20	[M-H]^−^	C_16_H_18_O_9_	353.0878	353.08798	0.50979	0.019	0.057
Catechin	3.27	[M-H]^−^	C_15_H_14_O_6_	289.07175	289.07205	1.03780	0.039	0.117
*p*-Coumaric acid	3.38	[M-H]^−^	C_9_H_8_O_3_	163.04001	163.03937	−3.92542	0.019	0.057
Apigenin-7-O-glucoside	3.48	[M-H]^−^	C_15_H_10_O_5_	269.04555	269.04526	−1.07788	0.019	0.057
Ferulic acid	3.55	[M-H]^−^	C_10_H_10_O_4_	193.05063	193.05016	−2.43459	0.039	0.117
Naringin	3.56	[M-H]^−^	C_27_H_32_O_14_	579.17193	579.17212	0.32805	0.019	0.057
Rutin	3.59	[M-H]^−^	C_27_H_30_O_16_	609.14611	609.14673	1.01782	0.019	0.057
Quercetin-3-galattoside	3.59	[M-H]^−^	C_21_H_20_O_12_	463.0882	463.08817	−0.06478	0.039	0.117
Hesperidin	3.61	[M-H]^−^	C_27_H_30_O_16_	609.14611	609.14612	0.01642	0.019	0.057
Kaempferol-3-glucoside	3.63	[M-H]^−^	C_21_H_20_O_11_	447.09195	447.09329	2.99715	0.019	0.057
Genistein	3.68	[M-H]^−^	C_15_H_10_O_5_	269.04554	269.04562	0.29735	0.019	0.057
Isorhamnetin-3-rutinoside	3.72	[M-H]^−^	C_28_H_32_O_16_	623.16176	623.16174	−0.03209	0.019	0.057
Myricetin	3.73	[M-H]^−^	C_14_H_10_O8	317.03029	317.02924	−3.31199	0.019	0.057
Daidzein	3.77	[M-H]^−^	C_15_10_0_O_4_	253.05063	253.05035	−1.10650	0.019	0.057
Quercetin	3.87	[M-H]^−^	C_15_H_10_O_7_	301.03538	301.03508	−0.99656	0.019	0.057
Naringenin	3.90	[M-H]^−^	C_15_H_12_O_5_	271.0612	271.0611	−0.36892	0.019	0.057
Luteolin	3.96	[M-H]^−^	C_15_H_10_O_6_	285.04046	285.04086	1.40331	0.039	0.117
Apigenin	4.07	[M-H]^−^	C_15_H_10_O_5_	269.04555	269.04556	0.03717	0.039	0.117

LOD: limit of detection; LOQ: limit of quantification.

**Table 2 antioxidants-11-02453-t002:** Flavonoids and phenolic acids content in broad bean hull extracts.

Compounds	Mean (mg/100 g)	±SD
PHENOLIC ACIDS		
*Benzoic Acids*		
Protocatechuic acid	ND	
Gallic acid	4.6	0.3
SUM	4.6	0.2
*Cinnamic acids*		
Ferulic acid	19.6	
Quinic acid	4.1	0.4
Chlorogenic acid	32.6	1.8
*p*-Coumaric acid	42.1	1.3
SUM	98.4	0.7
FLAVONOIDS		
*Flavanols*		
Epicatechin	31.2	1.2
Catechin	12.1	0.8
SUM	43.3	0.3
*Flavones*		
Kaemferol 3-glucoside	4.1	0.2
Apigenin-7-O-glucoside	ND	
Apigenin	0.6	0.1
Luteolin	0.5	0.1
SUM	5.2	0.1
*Flavonols*		
Quercetin	3.1	0.2
Isorhamnetin-3-rutinoside	ND	
Quercetin-3-galattoside	0.8	0.1
Rutin	12.3	1.1
SUM	16.2	0.6
*Flavanones*		
Hesperidin	15.4	1.2
Naringenin	2.8	0.1
Naringin	ND	
SUM	18.2	0.8
*Isoflavone*		
Daidzein	ND	
Myricetin	9.2	0.2
Genistein	3.1	0.1
SUM	12.3	0.3
Total Polyphenols	198.2	0.3

**Table 3 antioxidants-11-02453-t003:** Total phenolic content measured in encapsulated and non-encapsulated broad bean hull extract.

Samples	TPC mg GAE/g ± SD	
Not Digested Extract	9.7 ± 0.4	
	Extract	Capsule
Digestion Stage		
Intestinal stage	2.7 ± 0.2 *	3.4 ± 0.1 *
Pronase	3.5 ± 0.3 *	4.3 ± 0.3 *
Viscozyme L	3.1 ± 0.3 *	3.9 ± 0.2 *
Total colonic stage	6.6 ± 0.3 *	8.2 ± 0.3 *

Differences between groups were statistically analyzed with Tukey’s test; * *p*-value ≤ 0.05 extract versus capsule.

**Table 4 antioxidants-11-02453-t004:** Antioxidant capacity measured in broad bean hull extract encapsulated and non-encapsulated evaluated by DPPH and ABTS tests.

	DPPH mmol/kg ± SD		ABTS mmol/kg ± SD	
Not Digested Extract	15.2 ± 1.3		19.1 ± 1.4	
	**Extract**	**Capsule**	**Extract**	**Capsule**
Digestion stage				
Intestinal stage	2.7 ± 0.2 *	3.1 ± 0.2 *	3.5 ± 0.3 *	4.2 ± 0.3 *
Pronase stage	2.3 ± 0.2 *	4.1 ± 0.4 *	4.1 ± 0.2 *	6.1 ± 0.3 *
Viscozyme L stage	2.1 ± 0.3	2.3 ± 0.1	3.2 ± 0.3 *	4.2 ± 0.5 *
Total colonic stage	4.4 ± 0.3 *	6.4 ± 0.3 *	7.3 ± 0.3 *	10.3 ± 0.4 *

Differences between groups were statistically analyzed with Tukey’s test; * *p*-value ≤ 0.05 extract versus capsule.

## Data Availability

Data is contained within the article and Appendix A.

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
