# Peer review of "Analysis of Polyphenolic Compounds in Water-Based Extracts of Vicia faba L.: A Potential Innovative Source of Nutraceutical Ingredients"

_antioxidants, 2022, doi:10.3390/antiox11122453_

Round 1

Reviewer 1 Report

The manuscript “Analysis of Polyphenolic Compounds in Water-Based Extractsof Vicia faba L.: A Potential Innovative Source of Nutraceutical Ingredients” investigated the polyphenolic profile of an aqueous extract from the hull of beans and evaluated the stability of total phenolics and in vitro antioxidant capacity over gastrointestinal digestion.

There are three main aspects that the authors should resolve for considering this manuscript for publication:

-        The authors must analyze the phenolic profile of the encapsulated and non-encapsulated (control) extract over gastrointestinal digestion (not just TPC).

-        The antioxidant capacity must be measured using a more physiological approach (cell culture experiments or in vivo assays).

-        The authors should statistically analyze all the information provided and discuss the results accordingly.

Author Response

Reviewer 1

The manuscript “Analysis of Polyphenolic Compounds in Water-Based Extractsof Vicia faba L.: A Potential Innovative Source of Nutraceutical Ingredients” investigated the polyphenolic profile of an aqueous extract from the hull of beans and evaluated the stability of total phenolics and in vitro antioxidant capacity over gastrointestinal digestion.

 There are three main aspects that the authors should resolve for considering this manuscript for publication:

-        The authors must analyze the phenolic profile of the encapsulated and non-encapsulated (control) extract over gastrointestinal digestion (not just TPC).

The characterization of the phenolic profile of the encapsulated and non-encapsulated (control) extract over gastrointestinal digestion will be performed in future work that we intend to carry out. The authors explained in the text the preliminary nature of this work and the fact that more studies are needed. The authors proceeded to add the suggested information in the conclusions section and reported as: “However, further research is needed to expand the understanding regarding the bioactivity of the studied extracts for possible applications as ingredients in nutraceutical formulations and to elucidate the biotransformation of the polyphenolic compounds that occurred during the GI process."

 -        The antioxidant capacity must be measured using a more physiological approach (cell culture experiments or in vivo assays).

As suggested by Reviewer 1, cell culture experiments and in vivo assays are the most common technique for assessing both the antioxidant capacity and polyphenol bioaccessibility. The authors proceeded to add the suggested information in the discussion section as: Several methods have been proposed to evaluate the antioxidant capacity after simulated GI digestion. The use of immortalized cell lines such as Caco-2 represents the most common technique for assessing antioxidant capacity and polyphenol bioaccessibility. However, to obtain an overview of antioxidant capacity, in the present work DPPH and ABTS tests were measured after GI process. The data highlighted strong correlations among the results obtained from TPC by the Folin-Ciocalteu and antioxidants tests assessed during in vitro GI process, supporting that the assayed methods provide some reliable information on the antioxidant compounds released by the nutraceutical formulations assayed during the simulated GI digestion”

 -        The authors should statistically analyze all the information provided and discuss the results accordingly.

As suggested by Reviewer 1, the authors added the missing information.

The authors thank the Reviewer 1 for evaluating our manuscript.

Reviewer 2 Report

I believe that the manuscript is clear and well organised. Authors give the research question and fundamental purpose of this study. As a whole, the information presented in this manuscript is a useful addition to scientific knowledge. So I suggest the publication in Journal of Antioxidants after some minor revision.

The followings are some minors comments and suggestions for author to improve the manuscript.

The results given in Tables 2 and 3 need statistical analysis

Table S2 could be presented in the main text

Author Response

Reviewer 2

I believe that the manuscript is clear and well organised. Authors give the research question and fundamental purpose of this study. As a whole, the information presented in this manuscript is a useful addition to scientific knowledge. So I suggest the publication in Journal of Antioxidants after some minor revision.

The followings are some minors comments and suggestions for author to improve the manuscript.

The results given in Tables 2 and 3 need statistical analysis

As suggested by Reviewer 2, the authors added the missing information.

Table S2 could be presented in the main text

As suggested by Reviewer 2, Table S2 is presented in the main text

The authors thank the Reviewer 2 for evaluating our manuscript.

Reviewer 3 Report

Dear Authors,

Your manuscript "Analysis of Polyphenolic Compounds in Water-Based Extracts of Vicia faba L.: A Potential Innovative Source of Nutraceutical Ingredients" is very interesting and original. In my opinion, this manuscript would be of interest not only to readers of the Journal of Antioxidants. The goal that the authors have set for themselves with this manuscript is not only research, but also to be used as a food supplement.

In my opinion, the Introduction to the authors part is well written and provides comprehensive information about the purpose the authors have set for themselves. Many of the cited literary sources are from the last 3-5 years, which shows that the authors were very serious about the goal they set themselves. I would also like to note that I also found a self-citation.

In the Methods and Materials section, the authors have well described the methods they used. In my opinion, any researcher could repeat them. I think it is correct according to UPAC to write the compound MeOH as CH3OH.

Results are presented accurately and clearly. The discussion corresponds to the results presented and is supported by relevant quotations. In my opinion, the goal that the authors set for themselves with this investigation of the polyphenolic profile of the water-based extract of bean husks using UHPLC-Q-Orbitrap is very serious and thorough. With the help of their obtained results and the corresponding discussion, the authors manage to prove that this study is done for the first time using water as an extracting agent or also called "green extraction". These methods, in contrast to various organic solvents, protect the environment and that the resulting water extract from the bean shells can be considered a valuable source of natural polyphenolic compounds.

In my opinion, the conclusion supports the results obtained by the authors.

I would also like to mention about the References part that out of 39 sources, 28 are from the last 3-5 years, but unfortunately I found that there are also a few self-citations.

Author Response

Reviewer 3

Dear Authors,

Your manuscript "Analysis of Polyphenolic Compounds in Water-Based Extracts of Vicia faba L.: A Potential Innovative Source of Nutraceutical Ingredients" is very interesting and original. In my opinion, this manuscript would be of interest not only to readers of the Journal of Antioxidants. The goal that the authors have set for themselves with this manuscript is not only research, but also to be used as a food supplement.

In my opinion, the Introduction to the authors part is well written and provides comprehensive information about the purpose the authors have set for themselves. Many of the cited literary sources are from the last 3-5 years, which shows that the authors were very serious about the goal they set themselves. I would also like to note that I also found a self-citation.

In the Methods and Materials section, the authors have well described the methods they used. In my opinion, any researcher could repeat them. I think it is correct according to UPAC to write the compound MeOH as CH3OH.

As suggested by Reviewer 3, the authors removed the word “MeOH” in the manuscript.

Results are presented accurately and clearly. The discussion corresponds to the results presented and is supported by relevant quotations. In my opinion, the goal that the authors set for themselves with this investigation of the polyphenolic profile of the water-based extract of bean husks using UHPLC-Q-Orbitrap is very serious and thorough. With the help of their obtained results and the corresponding discussion, the authors manage to prove that this study is done for the first time using water as an extracting agent or also called "green extraction". These methods, in contrast to various organic solvents, protect the environment and that the resulting water extract from the bean shells can be considered a valuable source of natural polyphenolic compounds.

In my opinion, the conclusion supports the results obtained by the authors.

I would also like to mention about the References part that out of 39 sources, 28 are from the last 3-5 years, but unfortunately I found that there are also a few self-citations.

In our point of view, there are legitimate reasons for these self-citations. In recent years my group and I have been involved in the valorization of agri-food waste and by-products, investigating the possible recovery of bioactive compounds to be used as innovative ingredients in the formulation of nutraceuticals, the same purpose of the present study. In our previous works, we have proposed several protocols and obtained a wide range of results that have been useful in the present scientific work.

The authors thank the Reviewer 3 for evaluating our manuscript.